# Physical Fitness and Physical Activity in Young Adults: A Comparative Study Between Two Higher Education Institutions

**DOI:** 10.3390/jfmk11010022

**Published:** 2025-12-31

**Authors:** Tatiana Sampaio, João P. Oliveira, Pedro M. Magalhães, José A. Bragada, Raul F. Bartolomeu, Catarina Vasques, Jorge E. Morais

**Affiliations:** 1Department of Sports Sciences, University of Beira Interior, 6201-001 Covilhã, Portugal; tatiana_sampaio30@hotmail.com (T.S.); jpco-2001@live.com.pt (J.P.O.); 2Research Centre in Sports Sciences, Health Sciences and Human Development (CIDESD), 5000-801 Vila Real, Portugal; 3Research Centre for Active Living and Wellbeing (LiveWell), Instituto Politécnico de Bragança, 5300-253 Bragança, Portugal; pmaga@ipb.pt (P.M.M.); jbragada@ipb.pt (J.A.B.); rbartolomeu@ipb.pt (R.F.B.); catarinav@ipb.pt (C.V.); 4Department of Sport Sciences, Instituto Politécnico de Bragança, 5300-253 Bragança, Portugal; 5Department of Sports and Expressions, Polytechnic of Guarda, 6300-559 Guarda, Portugal; 6Sport Physical Activity and Health & Innovation Center (SPRINT), 2040-413 Rio Maior, Portugal

**Keywords:** physical activity, cardiorespiratory fitness, sedentary behavior, young adults

## Abstract

**Background**: Evidence suggests that physical activity and fitness tend to decline during the transition to higher education, yet the influence of institutional environment on these variables remains unclear. **Objectives:** This study aimed to compare physical fitness and physical activity levels between first-year students from two higher education institutions within the same national context. **Methods**: Fifty-eight male university students (IPB (Instituto Politécnico de Bragança): *n* = 31; IPG (Instituto Politécnico da Guarda): *n* = 27; mean age IPB = 19.2 ± 1.8 years; IPG = 19.8 ± 5.5 years) were assessed for body composition, handgrip strength, mid-thigh pull, standing long jump, stork balance, flexibility, and estimated VO_2max_ (StepTest4all). Physical activity levels were determined using the International Physical Activity Questionnaire (IPAQ) and were classified as low, moderate, or high. Between-group comparisons were performed using independent samples *t*-tests, with statistical significance set at *p* < 0.05. **Results**: Students from IPB presented significantly higher IPAQ scores (2.97 ± 0.18) compared with those from IPG (2.56 ± 0.64; t = 3.235, *p* = 0.003, d = 0.90), along with superior standing long jump performance (1.95 ± 0.15 m vs. 2.12 ± 0.24 m; t = −3.239, *p* = 0.002, d = 0.85). No significant differences were observed for body composition, strength, flexibility, balance, psychological well-being or VO_2max_ (all *p* > 0.05), although small effects were noted for flexibility (d = 0.50) and VO_2max_ (d = 0.48). The distribution of physical activity categories revealed that 96.8% of IPB students were classified as highly active, whereas IPG students were more evenly distributed across high (63.0%), moderate (28.6%), and low (7.4%) activity levels. **Conclusions**: These findings indicate that institutional environment and access to exercise opportunities may influence physical activity behavior and lower-limb power in university students. Promoting structured physical activity programs and recreational opportunities within higher education may help sustain adequate fitness and health in this population.

## 1. Introduction

Regular engagement in physical activity is a recognized determinant of health and longevity, influencing metabolic, cardiovascular, and functional outcomes across the lifespan [1,2,3,4]. Despite strong scientific evidence supporting its benefits, a large proportion of the global population remains insufficiently active, which has made physical inactivity one of the most pressing public health challenges worldwide [5,6,7]. The limited participation in exercise and the growing prevalence of sedentary lifestyles contribute to the early development of non-communicable diseases, even among young adults and students [8].

University students represent a population group particularly vulnerable to reductions in daily movement. The transition into higher education often coincides with increased academic workload, stress, and social obligations, which collectively reduce opportunities for exercise [9,10]. This transition period is frequently associated with behavioral changes in physical activity patterns. Evidence suggests that students who were physically active prior to university may experience a decline in activity levels due to the loss of organized sport participation, reduced parental and institutional support, and greater autonomy over lifestyle choices. Conversely, students who were already inactive tend to maintain sedentary behaviors during higher education, as academic demands, prolonged screen time, and limited perceived competence or motivation act as barriers to engagement in physical activity [9,10]. Together, these factors contribute to the consolidation of physical inactivity and sedentary behavior during university years, highlighting this stage as a critical period for long-term lifestyle behavior formation.

Recent epidemiological data indicate that a substantial proportion of university students do not meet the World Health Organization recommendations for physical activity, with high levels of sedentary behavior reported across different countries [8]. Several studies have shown that this period is associated with a measurable decline in both physical activity and fitness [11,12,13]. Such trends are concerning because the behavioral patterns established during young adulthood tend to persist later in life, affecting long-term health trajectories.

Physical fitness is a multifactorial construct encompassing strength, endurance, flexibility, balance, and body composition, all of which are essential for maintaining functional health [14]. However, university students frequently exhibit deficits across these domains. Caia et al. [15] reported that more than half of the assessed students presented low strength, and Kubieva et al. [16] observed problems related to body mass and muscular condition independent of their activity level. Similar results have been documented for cardiorespiratory fitness, which can be improved only through consistent lifestyle modification [17,18].

Evidence also indicates that the determinants of fitness in this population extend beyond individual motivation. Cultural background, academic curriculum, and institutional policies play significant roles. For instance, students enrolled in faculties of sport or physical education usually display higher fitness levels compared to peers from other disciplines, mainly due to the inclusion of practical exercise components in their programs [19,20,21]. Conversely, students from non-sport faculties often demonstrate moderate or low levels of physical activity [19,22]. Sex differences have also been observed, with males typically outperforming females in strength- and power-related tests, whereas females tend to achieve better flexibility scores [23].

These findings suggest that the variability in physical activity and fitness among university students cannot be attributed to a single factor. Differences across studies may reflect distinct cultural contexts, educational systems, or access to physical activity facilities [19,24]. While international comparisons have highlighted such disparities, fewer studies have examined how these differences manifest within a single country, where institutional environments, teaching approaches, and resources may still vary considerably.

Given this background, examining the fitness and activity profiles of university students in comparable national settings can provide valuable insights into how institutional conditions influence health-related behaviors. Current evidence on physical activity and physical fitness in university students has predominantly focused on international comparisons, specific academic programs (e.g., sport sciences versus non-sport curricula), or single-institution samples [8]. Although these studies consistently report insufficient physical activity levels and heterogeneous fitness profiles, less attention has been given to how institutional context within the same national and educational framework may influence students’ physical activity behaviors and physical fitness outcomes [19]. By comparing first-year students from two higher education institutions within the same country using a comprehensive and standardized fitness assessment battery, the present study contributes to the existing literature by highlighting potential institutional influences on physical activity patterns and selected components of physical fitness in young adults.

Therefore, the aim of this study was to compare physical activity levels and multiple components of physical fitness as well as psychological well-being among first-year university students from two higher education institutions within the same national context. Based on the existing literature, it was hypothesized that first-year university students from different higher education institutions would present distinct physical activity levels. This comparison seeks to identify potential differences associated with institutional environments, local contexts, and physical activity promotion strategies.

## 2. Materials and Methods

### 2.1. Participants

This study followed a cross-sectional observational design and was conducted at two Portuguese higher education institutions: the Instituto Politécnico de Bragança and the Instituto Politécnico da Guarda. The sample comprised 58 male university students with an average age of 19.7 ± 2.5 years, distributed between two institutions: IPB (*n* = 31) and IPG (*n* = 27). The details per institution are presented in Table 1. The inclusion criteria required participants to be first-year students enrolled at each institution and free from any injury that could interfere with testing procedures. Exclusion criteria included the use of medication known to influence physical performance or physiological responses, as well as pregnancy. All procedures were conducted in accordance with the Declaration of Helsinki for research involving human participants, and the study protocol received approval from the Institutional Ethics Committee (approval number: P533182-R654085-D1985073).

### 2.2. Research Design

Data collection was carried out at both institutions (IPB and IPG) during the first week of classes, across three non-consecutive days. When working with relatively large samples, selecting simple, quick, and reliable measures is essential to ensure efficiency and consistency while maintaining methodological rigor. The selected variables fulfilled these criteria. On the first day, participants completed the short version of the International Physical Activity Questionnaire (IPAQ-short) [25] as well as the Well-being Questionnaire [26], and a set of anthropometric measurements was collected. On the second day, cardiorespiratory fitness was assessed using the StepTest4all protocol [27], a standardized and feasible procedure for estimating maximal oxygen uptake (VO_2max_) through a progressive step test. On the third day, tests related to muscular strength, flexibility, and balance were performed. The mid-thigh pull (MTP) test assessed maximal isometric strength of the lower and total body [28]. Handgrip strength (HG) evaluated upper-limb strength, serving as a reliable indicator of general muscular functionality [29]. The standing long jump (SLJ) test measured lower-limb explosive power [30]. Flexibility was assessed with the sit-and-reach (S&R) test, which reflects hamstring and lower-back flexibility and general mobility [31]. Finally, static balance was evaluated through the standing stork test (SST) [32] (see Figure 1).

### 2.3. Data Collection

#### 2.3.1. Physical Activity and Well-Being Questionnaire

The short version of the International Physical Activity Questionnaire (IPAQ-short) was used to assess participants’ physical activity levels. This instrument classifies individuals into three categories of physical activity (PA): low, moderate, and high [33]. The scoring process is based on the total weekly PA volume, calculated by multiplying the frequency (days per week), duration (minutes per day), and the corresponding metabolic equivalent (MET) value for each activity intensity [25]. According to the official IPAQ scoring protocol, participants are classified as: (i) low—individuals whose activity does not reach the minimum criteria for moderate classification; (ii) moderate—those performing at least three days of vigorous activity for ≥20 min per day, or five or more days of moderate, vigorous, or walking activities for ≥30 min per day, accumulating at least 600 MET·min/week; and (iii) high—those engaging in vigorous activity on at least three days, achieving ≥1500 MET·min/week, or completing seven or more days of any combination of activities (walking, moderate, or vigorous) with a total energy expenditure of ≥3000 MET·min/week.

Psychological well-being was assessed using the reduced version of the *Escala de Bem-Estar Psicológico* (EBEP). This questionnaire is based on the original Psychological Well-Being Scales developed by Ryff (1989) [34] and later translated and adapted for the Portuguese population. The reduced 18-item version follows the structure described by Fernandes, Vasconcelos-Raposo, and Teixeira [26] and evaluates six dimensions of psychological well-being: autonomy, environmental mastery, personal growth, positive relations with others, purpose in life, and self-acceptance. Each item is rated on a six-point Likert scale ranging from “strongly disagree” to “strongly agree”. Eight items are positively worded and ten are negatively worded, and scoring follows the recommendations for corrected item attribution. In the present study, the EBEP score was used only as descriptive information to complement the characterization of the sample.

#### 2.3.2. Anthropometrics

Height was measured using a digital stadiometer (SECA 242, Hamburg, Germany). Subsequently, body mass (BM, kg), body mass index (BMI, kg/m^2^), fat mass percentage (%FM), and lean mass percentage (%LM) were assessed using a bioelectrical impedance analyzer (Tanita MC-780P MA, Tokyo, Japan) [35], following the manufacturer’s standard operating procedures. All anthropometric and body composition assessments were performed in the morning. Participants were instructed to attend the measurements in a fasted state, to avoid strenuous physical activity in the previous 24 h, and to maintain normal hydration.

#### 2.3.3. Cardiorespiratory Fitness

Cardiorespiratory fitness was assessed using the StepTest4all protocol [27], a standardized procedure developed to estimate maximal oxygen uptake (VO_2max_) through a progressive step test. The test involves stepping on and off a platform at a controlled pace for a set duration, allowing for an indirect evaluation of aerobic capacity and cardiovascular endurance. In the present study, a step height of 40 cm and a fast cadence were used, reaching approximately 80% of the estimated maximal heart rate (HR_max_) within 5 to 10 min. The test was terminated when one of the following criteria was met: (i) HR reached 80% of HR_max_, (ii) the participant reported discomfort or fatigue, or (iii) the participant failed to maintain the required pace. All participants achieved the 80% HR_max_ threshold. Heart rate was continuously monitored during the test and recovery using a Garmin chest strap (Garmin International Inc., Olathe, KS, USA). All assessments were performed with participants wearing appropriate fitness clothing and shoes. The HRmax was estimated as:(1)$${HR}_{max}=208\cdot age$$
where HR_max_ is the maximal heart rate (in bpm) and age is the chronological age (in years) [36]. As for the VO_2max_, this was estimated as:(2)$${VO}_{2max}=17.105+0.260\cdot {HRR}_{60}+8.563\cdot \left(sex\right)+4.097\cdot ({PA}_{level})$$
where VO_2max_ is the maximum oxygen uptake (in mL·kg^−1^·min^−1^), HRR_60_ is the heart rate recovery during 60 s (in bpm) immediately after the end of the step test, sex is zero for women and 1 for men, and PA_level_ is the physical activity level (level 1—low; level 2—moderate; level 3—high) [27].

#### 2.3.4. Muscular Strength, Balance and Flexibility

Participants wore appropriate fitness clothing and shoes for the assessments of muscular strength, balance, and flexibility. Before testing, they completed a standardized 5–10 min warm-up that included light jogging, jumping jacks, skipping, dynamic stretching, and mobility exercises. All participants were thoroughly familiarized with each test and allowed to perform a brief trial before data collection.

For the mid-thigh pull (MTP) test, maximal isometric strength (in kg) was measured using a digital dynamometer (Takei T.K.K. 5402, Takai, Japan). The bar height was individually adjusted so that the handles aligned with each participant’s mid-thigh level, ensuring optimal joint positioning, approximately 125°–145° at the knee and 140°–150° at the hip [28]. Participants were instructed to adopt an upright posture with a slight knee flexion, dorsiflexed ankles, retracted and depressed shoulders, and feet positioned hip-width apart beneath the handle [37]. After achieving a stable position, a countdown was given, and participants performed a maximal pull, applying force as rapidly and forcefully as possible while simultaneously pushing their feet into the ground. Minimal pre-tension was allowed to avoid slack before the pull [38]. Each participant completed three maximal five-second trials, interspersed with two-minute rest periods, and the highest value was used for further analysis.

Handgrip strength (HG) was assessed in the dominant upper limb, with dominance determined by self-report. A digital hand dynamometer (JAMAR, Lafayette, LA, USA) was used to measure maximal isometric grip strength (kg). Participants stood in an upright position with both arms fully extended along the trunk and were instructed to exert maximal force with their dominant hand [39]. Three maximal attempts were performed, each separated by a two-minute rest interval to ensure full recovery, and the highest value was retained for analysis.

The standing long jump (SLJ) test was used to assess lower-limb explosive strength [30]. Participants stood behind a marked line with their feet shoulder-width apart and were instructed to flex their knees, swing their arms backward, and jump forward as far as possible in one continuous motion, landing with both feet parallel. An experienced evaluator measured the jump distance (in cm) from the take-off line to the nearest heel using a measuring tape (RossCraft, Canada). Each participant performed three trials, and the best result was retained for analysis. The sit-and-reach (S&R) test was used to assess flexibility of the lower back and hamstring muscles [40]. A standard box with a sliding ruler was employed, with the 0 cm mark aligned with the participants’ toe level. Negative values indicated that participants could not reach the toes, whereas positive values reflected the ability to extend beyond them. Scores were recorded to the nearest 0.5 cm. Based on [41], performance was classified as: (i) excellent—>14 cm for males and >15 cm for females; (ii) above average—11.0–14.0 cm for males and 12.0–15.0 cm for females; (iii) average—7.0–10.9 cm for males and 7.0–11.9 cm for females; (iv) below average—4.0–6.9 cm for both sexes; and (v) poor—<4.0 cm for both sexes. Each participant performed three attempts, and the best score was used for analysis.

The standing stork test (SST) was used to assess static balance [42]. Participants stood barefoot on a flat surface with their hands placed on their hips. Each participant selected their dominant leg as the supporting limb and raised the opposite foot, placing the toes of the lifted foot against the inside of the knee of the standing leg. Once stable, the timer was started using a stopwatch. Participants were instructed to maintain balance for as long as possible in this position. The test ended if (i) the raised foot lost contact with the supporting leg, (ii) the heel of the supporting foot touched the ground, (iii) the hands were removed from the hips, or (iv) balance was lost. Three trials were performed, and the longest duration was retained for analysis. Normative reference values were as follows: males—excellent >50 s, above average 37–50 s, average 15–36 s, below average 5–14 s, poor <5 s; females—excellent >27 s, above average 23–27 s, average 8–22 s, below average 3–7 s, poor <3 s.

### 2.4. Statistical Analysis

Tests of data distribution (Shapiro–Wilk) and homogeneity (Levene’s test) were conducted before analysis. The data are shown in descriptive statistics for means and one standard deviation. Descriptive statistics were presented as mean ± standard deviation (SD). Independent samples *t*-tests were conducted to compare the two institutions (IPB vs. IPG) across all variables. Mean differences with 95% confidence intervals (95% CI) were computed, and statistical significance was set at α = 0.05. Cohen’s d was used to estimate the standardized effect size between pairwise: (i) trivial if 0 ≤ d < 0.20; (ii) small if 0.20 ≤ d < 0.60; (iii) moderate if 0.60 ≤ d < 1.20; (iv) large if 1.20 ≤ d < 2.00; (v) very large if 2.00 ≤ d < 4.00, and; (vi) perfectly distinct if d ≥ 4.00 [43]. In addition, categorical variables representing physical activity level and VO_2max_ classification were analyzed using frequency and percentage distributions. Data were analyzed using SPSS, v.29.0 for Windows (SPSS Inc., Chicago, IL, USA).

## 3. Results

Table 1 presents the descriptive statistics for anthropometric and physical fitness variables among first-year university students from the two higher education institutions. This table provides an overview of the participants’ general characteristics, including body composition, stature, and fitness-related measures

The between-group comparisons for all performance and fitness variables are presented in Table 2, with an additional graphical representation shown in Figure 2. Significant differences between institutions were observed for two variables. The IPAQ score was significantly higher in students from IPB compared to IPG (mean difference = 0.41, 95% CI = 0.15 to 0.67, t = 3.235, *p* = 0.003, d = 0.90, moderate effect). Similarly, the standing long jump performance was superior in IPB students (mean difference = −0.17 m, 95% CI = −0.28 to −0.06, t = −3.239, *p* = 0.002, d = 0.85, moderate effect).

No statistically significant differences were found between institutions for body mass, height, BMI, fat mass, lean mass, grip strength, mid-thigh pull, stork test, flexibility, or VO_2max_ (all *p* > 0.05). Nevertheless, small effect sizes were observed in body mass (d = 0.42), BMI (d = 0.36), stork test (d = 0.40), flexibility (d = 0.50), and VO_2max_ (d = 0.48), indicating minor but potentially meaningful practical differences (Table 2).

Figure 3 illustrates the distribution of participants according to physical activity level (Panel A) and VO_2max_ classification (Panel B) for each institution. In terms of physical activity level (IPAQ), almost all students from IPB were classified as highly active (96.8%, *n* = 30), with only one student in the moderate category (3.2%) and none in the low category. In contrast, IPG students showed a more heterogeneous profile, with 63.0% classified as highly active (*n* = 17), 28.6% as moderate (*n* = 8), and 7.4% as low activity (*n* = 2). Regarding cardiorespiratory fitness (VO_2max_), IPB students were mostly distributed across the fair (51.6%, *n* = 16) and poor (22.6%, *n* = 7) categories, with fewer students classified as good (19.4%, *n* = 6), and only two students in the very poor category (6.5%). No IPB student achieved an “excellent” VO_2max_ level. Conversely, IPG participants showed a wider spread across fitness categories, with 44.4% classified as fair (*n* = 12), 18.5% as poor (*n* = 5), 18.5% as good (*n* = 5), and 18.5% as excellent (*n* = 5), while none were classified as very poor.

The psychological well-being score showed similar values between institutions. Students from IPB presented a mean score of 80.23 ± 10.50, while students from IPG scored 75.31 ± 14.01. The difference between groups was not statistically significant (t(55) = 1.51, *p* = 0.136), although the effect size was small (d = 0.40). These results suggest that both groups reported generally high levels of perceived well-being, with only minor differences that do not meaningfully influence the main findings of the study.

## 4. Discussion

This study compared physical fitness and physical activity profiles among first-year university students from two higher education institutions within the same national context. The main findings revealed that students from IPB presented significantly higher IPAQ scores and superior standing long jump performance compared with those from IPG. No significant differences were observed for cardiorespiratory fitness, muscular strength, flexibility, or balance. These results indicate that institutional factors may influence the expression of specific fitness attributes and daily activity behaviors, even when populations share similar demographic and educational characteristics.

The findings of the present study align with the broader trends reported in the systematic review by Kljajević et al. [8], which synthesized evidence on physical activity and fitness among university students worldwide. That review concluded that students generally display satisfactory but heterogeneous fitness and activity levels, largely influenced by contextual factors such as study program, institutional environment, and national culture. Specifically, it highlighted that university students often experience a decline in physical activity compared with their pre-university years, and that participation in structured exercise or sport programs is a key determinant of maintaining adequate fitness levels. These observations reinforce the interpretation that differences between institutions, even within the same educational system, may stem from environmental and organizational factors rather than individual characteristics. The higher activity levels reported by IPB students are likely influenced by environmental and institutional determinants. Although both institutions operate under comparable academic structures, differences in the local availability of sport facilities, extracurricular opportunities, and institutional support for active lifestyles may partially explain the observed discrepancies. Previous investigations have demonstrated that the inclusion of structured physical education components and sport-related curricula is associated with higher fitness and activity levels in university students [19,20,21]. Griban et al. [19] further emphasized that institutional environment and curricular organization are key mediators of student participation in physical exercise. The present findings are consistent with this evidence, supporting the notion that institutional culture and infrastructure represent relevant contributors to students’ activity profiles.

The superior standing long jump performance observed in IPB students reinforces this interpretation. Explosive lower-limb power is sensitive to habitual exercise intensity and frequency, serving as an indirect marker of neuromuscular adaptation to regular physical activity [23,44,45]. Given that no significant differences were observed in maximal strength measures, such as handgrip and mid-thigh pull, the improved jump performance in IPB students likely reflects higher exposure to dynamic physical activities rather than differences in absolute strength capacity. No significant differences in cardiorespiratory fitness were found between institutions, with both groups presenting VO_2max_ values within the expected range for moderately active young adults [46,47]. The mean values obtained (≈40–43 mL·kg^−1^·min^−1^) correspond to “fair-to-good” classifications, indicating an overall satisfactory aerobic profile. The lack of between-group differences despite the disparity in IPAQ classifications suggests that a considerable portion of reported activity may consist of low-to-moderate intensity exercise, insufficient to induce measurable improvements in VO_2max_. Similar observations have been documented in previous studies, in which self-reported physical activity levels were not consistently reflected in objective fitness outcomes [15,16,19]. Additionally, minor but consistent differences in flexibility and balance were also detected, with IPB students exhibiting slightly higher mean values. Although these differences did not reach statistical significance, they may indicate greater exposure to diversified motor experiences or recreational sport participation. Previous findings have shown that interventions combining aerobic, resistance, and coordination exercises can lead to significant improvements in flexibility and balance performance [44,48]. Thus, even small variations in movement diversity and activity type may yield functional benefits over time.

The observed differences between institutions may be partially explained by contextual and environmental factors that were not directly assessed in the present study. Although both institutions belong to the same national higher education system, differences in undergraduate course profiles, academic demands, and daily schedules may influence students’ opportunities and motivation for physical activity. Previous research has shown that the transition to university is often accompanied by reduced participation in structured physical activity, increased academic workload, and greater sedentary behavior [9]. In addition, institutional context and environmental characteristics may contribute to heterogeneous physical fitness profiles among university students [19]. These factors may help explain the variability observed in physical activity levels and selected physical fitness outcomes between institutions.

Overall, the findings of the present study extend the available literature by demonstrating that measurable differences in fitness and physical activity can occur within a single national and educational framework. While previous research has attributed such variability primarily to cultural or socioeconomic factors [19,24], the present results suggest that institutional environment and program structure may exert comparable influence. These outcomes reinforce the importance of promoting active institutional policies that ensure equitable access to exercise opportunities across higher education settings. From a practical standpoint, universities should consider implementing systematic strategies to foster physical activity engagement, including regular fitness monitoring, structured exercise programs, and improved accessibility to sports facilities. Targeting first-year students may be particularly relevant, as this period is often characterized by reductions in habitual activity. Although data collection was conducted after the most restrictive phases of the COVID-19 pandemic, it should be acknowledged that the pandemic period may have had a residual influence on physical activity behaviors among university students. Previous evidence has shown significant reductions in physical activity levels during the pandemic, with potential long-term behavioral consequences [49]. Therefore, pandemic-related contextual factors may act as confounders when interpreting physical activity patterns in post-pandemic cohorts.

This study is not without limitations. The sample was limited to two institutions and a limited number of participants, which constrains the generalization of results. Physical activity was assessed through self-report, which may overestimate actual participation. Future studies should integrate objective monitoring tools (e.g., accelerometers) and longitudinal designs to evaluate how activity patterns and physical fitness evolve throughout university years. In addition, cardiorespiratory fitness was estimated using a submaximal field-based protocol, which is subject to inherent limitations and potential measurement inaccuracy when compared with direct assessment. Expanding the analysis to include psychosocial variables, such as motivation, perceived competence, or academic stress, could also provide a more comprehensive understanding of factors influencing students’ physical activity and fitness.

## 5. Conclusions

In summary, this study compared the physical fitness, well-being and physical activity levels of first-year students from two higher education institutions within the same national framework. The results demonstrated significantly higher physical activity scores and superior lower-limb power among students from IPB, while no significant differences were identified in cardiorespiratory fitness, muscular strength, flexibility, or balance. These findings suggest that institutional environment and the local culture of physical activity may influence certain components of fitness, particularly those related to activity behavior and explosive performance. Overall, the data indicate that both student groups exhibit adequate yet suboptimal fitness profiles, reinforcing the need for continuous promotion of structured physical activity in higher education. Implementing institutional strategies that facilitate regular exercise, ensure access to sports facilities, and encourage participation across academic programs may contribute to the improvement of students’ physical fitness and long-term health. Future studies using larger samples, longitudinal designs, and objective monitoring tools are warranted to better understand how these differences evolve throughout the university years.

## Figures and Tables

**Figure 1 jfmk-11-00022-f001:**
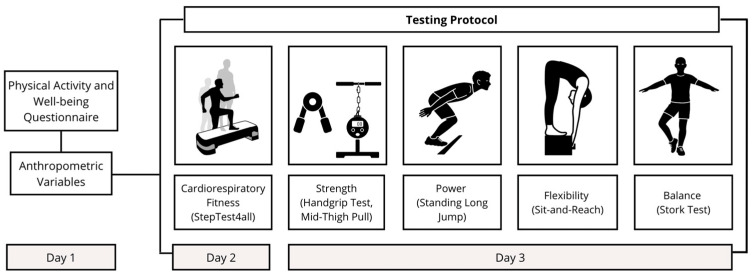
Schematic representation of the research design.

**Figure 2 jfmk-11-00022-f002:**
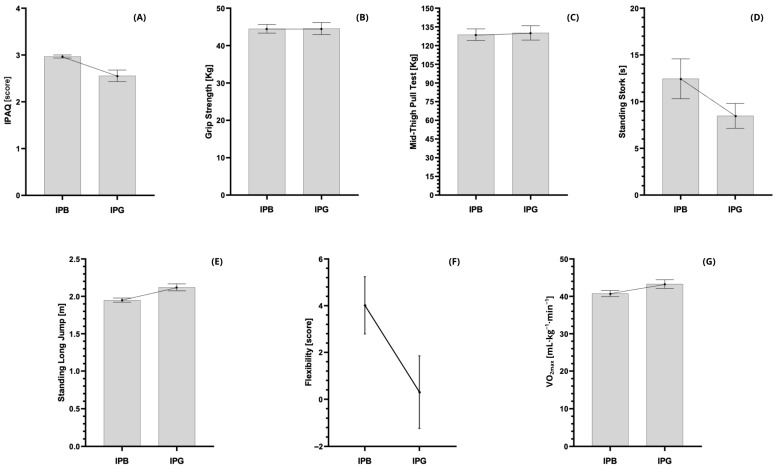
Comparison of physical performance and fitness outcomes between institutions. Panel (**A**): Physical activity level (IPAQ score); Panel (**B**): Grip strength; Panel (**C**): Mid-thigh pull; Panel (**D**): Standing stork balance test; Panel (**E**): Standing long jump; Panel (**F**): Sit-and-reach flexibility test; Panel (**G**): Estimated VO_2max_. Bars represent mean values, and error bars indicate standard error of the mean (SEM). IPB = Instituto Politécnico de Bragança; IPG = Instituto Politécnico da Guarda.

**Figure 3 jfmk-11-00022-f003:**
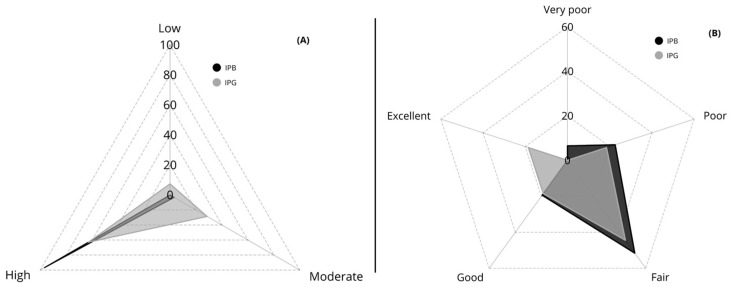
Distribution of participants according to (**A**) physical activity level (IPAQ) and (**B**) cardiorespiratory fitness classification (VO_2max_ levels) by institution. Values are expressed as percentages. IPAQ = International Physical Activity Questionnaire; VO_2max_ = Maximal oxygen uptake estimated through the StepTest4all protocol [27] IPB = Instituto Politécnico de Bragança; IPG = Instituto Politécnico da Guarda.

**Table 1 jfmk-11-00022-t001:** Descriptive statistics (mean ± standard deviation, SD) of the anthropometric and physical fitness variables of first-year university students from both institutions (IPB and IPG).

	IPB (*n* = 31)	IPG (*n* = 27)
	Mean ± SD	Mean ± SD
Age [Years]	19.19 ± 1.78	19.81 ± 5.53
Body mass [Kg]	70.70 ± 7.68	74.07 ± 8.40
Height [cm]	175.71 ± 5.70	176.15 ± 4.17
BMI [Kg/m^2^]	22.93 ± 2.55	23.90 ± 2.88

Note: BMI: Body Mass Index.

**Table 2 jfmk-11-00022-t002:** Descriptive statistics (mean ± standard deviation—SD) of all variables measured. It also presents information about the comparison between institutions (95CI—95% confidence intervals).

	IPB Mean ± SD	IPG Mean ± SD	Mean Difference (95 CI)	*t*-Test (*p*-Value)	Effect Size (Descriptor)
**Body Mass [Kg]**	70.70 ± 7.68	74.07 ± 8.40	−3.374 (−7.603 to 0.855)	−1.598 (0.116)	0.42 (small)
**Height [cm]**	175.71 ± 5.70	176.15 ± 4.17	−0.438 (−3.100 to 2.223)	−0.330 (0.743)	0.09 (trivial)
**BMI [Kg/m^2^]**	22.93 ± 2.55	23.90 ± 2.88	−0.968 (−2.396 to 0.461)	−1.357 (0.180)	0.36 (small)
**Fat Mass [%]**	16.16 ± 4.72	16.14 ± 6.24	0.027 (−2.862 to 2.917)	0.019 (0.985)	0.01 (trivial)
**Lean Mass [%]**	79.64 ±4.50	79.79 ± 6.16	−0.150 (−2.963 to 2.663)	−0.107 (0.915)	0.03 (trivial)
**IPAQ [Score]**	2.97 ± 0.18	2.56 ± 0.64	0.412 (0.152 to 0.673)	3.235 (0.003)	0.90 (moderate)
**Grip Strength [Kg]**	44.48 ± 6.41	44.56 ± 8.40	−0.072 (−3.975 to −3.832)	−0.037 (0.971)	0.01 (trivial)
**Mid-Thigh Pull [Kg]**	128.69 ± 26.09	130.15 ± 29.76	−1.455 (−16.141 to 13.232)	−0.198 (0.843)	0.05 (trivial)
**Standing Long Jump [m]**	1.95 ± 0.15	2.12 ± 0.24	−0.17 (−0.276 to −0.06)	−3.239 (0.002)	0.85 (moderate)
**Standing Stork Test [s]**	12.45 ± 11.87	8.47 ± 6.54	3.974 (−1.432 to 9.379)	1.475 (0.146)	0.40 (small)
**Flexibility [Score]**	4.02 ± 6.70	0.31 ± 8.02	3.706 (−0.206 to 7.617)	1.899 (0.063)	0.50 (small)
**Vo_2max_ [mL·kg^−1^·min^−1^]**	40.75 ± 4.51	43.27 ± 6.08	−2.519 (−5.313 to 0.275)	−1.806 (0.076)	0.48 (small)

Note: BMI: Body Mass Index; IPAQ: International Physical Activity Questionnaire. IPB = Instituto Politécnico de Bragança; IPG = Instituto Politécnico da Guarda.

## Data Availability

The data presented in this study are available on request from the corresponding author. The data are not publicly available due to privacy or ethical restrictions.

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
