# Peer review of "Physical Fitness and Physical Activity in Young Adults: A Comparative Study Between Two Higher Education Institutions"

_jfmk, 2025, doi:10.3390/jfmk11010022_

Round 1
Reviewer 1 Report
Comments and Suggestions for Authors
Thank you for the opportunity to read the paper entitled "Physical Fitness and Physical Activity in Young Adults: A Comparative Study between Two Higher Education Institutions". This is an interesting study with a comprehensive measurement of fitness from different sides (cardiorespiratory, muscular, etc.). However, some areas need improvement. Please see the detailed report below:
Abstract: Explain IPB and IPG abbreviations.
Abstract: You presented d, which is an effect size measure. But the effect size is always supported by some statistical tests, which you omitted in the brackets in the abstract. So that presents the statistical test results along with an effect size.
Materials and methods: This study looks like a cross-sectional one. So reschedule this paper according to STROBE guidelines and attach a checklist.
Materials and methods: Why did you use the Kolmogorov-Smirnov test (which is appropriate for larger samples)? You should use Shapiro-Wilk.
Materials and methods: Did you test the power of your sample?
Materials and methods: Specify the day (morning, evening, etc.) for measurements of height and body composition. Moreover, did you measure those things in the fasted state?
Discussion: You should acknowledge that the physical activity of students significantly deteriorated during the COVID-19 pandemic and that some particular periods should be considered as confounders. Recommended reference is doi: 10.3390/ijerph19148484.
Discussion: You estimated the VO2max of your subjects. You should acknowledge that the VO2max has certain limitations and poses some inaccuracy. Moreover, the gold standard for VO2max is CPET, which should be recommended for further studies.
Finally, the major revisions are recommended.
Author Response
Manuscript ID jfmk-4055618
Physical Fitness and Physical Activity in Young Adults: A Comparative Study between Two Higher Education Institutions
Reviewer #1
We would like to thank Reviewer 1 for the careful evaluation of our manuscript and for the constructive comments, which substantially improved the clarity and methodological rigor of the paper. All changes made to the manuscript are highlighted in the revised version.
Authors: Thank you very much for the time spent and the constructive feedback on this manuscript. We have made every effort to take on board your recommendations and comments. We hope this 2nd revised version and the responses to the comments (kindly refer to our replies below) will meet your requirements. Please note that all new changes in the revised manuscript are highlighted in yellow.
Abstract: Explain IPB and IPG abbreviations.
Authors: We appreciate the reviewer’s comment. The abbreviations IPB and IPG have now been fully defined at their first occurrence in the abstract as Instituto Politécnico de Bragança and Instituto Politécnico da Guarda, respectively.
Abstract: You presented d, which is an effect size measure. But the effect size is always supported by some statistical tests, which you omitted in the brackets in the abstract. So that presents the statistical test results along with an effect size.
Authors: We appreciate the reviewer’s comment. The abstract has been revised to include the corresponding statistical tests and p values alongside the reported effect sizes, ensuring a complete and transparent presentation of the results.
Materials and methods: This study looks like a cross-sectional one. So reschedule this paper according to STROBE guidelines and attach a checklist.
Authors: We appreciate the reviewer’s comment. The STROBE checklist has been added as supplementary material.
Materials and methods: Why did you use the Kolmogorov-Smirnov test (which is appropriate for larger samples)? You should use Shapiro-Wilk.
Authors: We appreciate the reviewer’s comment. The use of the Kolmogorov–Smirnov test in the original version was a reporting error. Normality was reassessed using the Shapiro–Wilk test, which is more appropriate for small to moderate sample sizes, and the manuscript has been corrected accordingly.
Materials and methods: Did you test the power of your sample?
Authors: We appreciate the reviewer’s comment. We acknowledge that an a priori or post hoc power analysis was not performed. This limitation has now been explicitly acknowledged in the limitations section of the manuscript, noting that the relatively small sample size may have reduced the statistical power to detect small between-group differences.
Materials and methods: Specify the day (morning, evening, etc.) for measurements of height and body composition. Moreover, did you measure those things in the fasted state?
Authors: We appreciate the reviewer’s comment. Anthropometric and body composition measurements were performed in the morning. Participants were instructed to attend the assessments in a fasted state and to avoid strenuous physical activity prior to testing, in accordance with the manufacturer’s recommendations for bioelectrical impedance analysis. This information has now been added to the Materials and Methods section.
Discussion: You should acknowledge that the physical activity of students significantly deteriorated during the COVID-19 pandemic and that some particular periods should be considered as confounders. Recommended reference is doi: 10.3390/ijerph19148484.
Authors: We appreciate the reviewer’s comment. We agree with the reviewer that the COVID-19 pandemic had a substantial impact on students’ physical activity levels. Although data collection in the present study occurred after the most restrictive pandemic period, we have now acknowledged in the discussion that pandemic-related behavioral changes may have persisted and should be considered as potential contextual confounders. The suggested reference has been added to the manuscript.
Discussion: You estimated the VO2max of your subjects. You should acknowledge that the VO2max has certain limitations and poses some inaccuracy. Moreover, the gold standard for VO2max is CPET, which should be recommended for further studies.
Authors: We appreciate the reviewer’s comment. This point has been addressed in the discussion. We explicitly acknowledge the limitations associated with estimating VOâ‚‚max through submaximal field tests.
Reviewer 2 Report
Comments and Suggestions for Authors
The manuscript entitled “Physical Fitness and Physical Activity in Young Adults: A Comparative Study between Two Higher Education Institutions” was reviewed. The article provides interesting information on the topic; however, adjustments need to be made so that the article can continue the path to publication. I kindly ask if all changes made to the text be highlighted in yellow or a different color in the text.
Below are the reviewer's considerations to be adjusted in the manuscript.
Abstract:
1- Add the topic “objectives” in the abstract before mentioning the objectives of the study.
2- What do the acronyms “IPB” and “IPG” mean in line 21 of the abstract?
3- Did you separate people into groups and when you presented their ages, did you average them together? Please maintain the standard and present the information separately by institution. In addition, take the opportunity to present the participants’ other sociodemographic and body composition information.
4- In the methods part of the abstract, please add information about the statistical tests and the significance index.
5- What were the IPAQ scores considered to compare the groups?
6- Present information on the mean and standard deviation in the results in addition to the analysis of statistical tests.
Keywords:
7- Avoid repeating words already mentioned in the title.
Introduction:
8- Present epidemiological information about sedentary behavior, physical inactivity and physical capacity in this student profile.
9- Mention in more detail what factors lead to this behavior in higher education. Do active students become physically inactive and inactive students remain inactive?
10- Detail the current state of the art on the topic and how this study will contribute to the evolution of the area.
11- What are the main hypotheses of the research? Please present more clearly.
12- Align the objectives presented at the end of the introduction with those mentioned in the abstract.
Materials and Methods:
13- Start the description of the methodology by addressing the type of study and the location where the research was carried out.
14- Present information about the sample calculation to justify the number of people selected.
15- Studies involving human beings have been suggested to register on the clinical trial platform, even if this is not the design of the study.
16- Prepare a flowchart of the study so that the reader can know how many people were recruited and how many remained for statistical analyses.
17- You need to reorganize figure 1 according to the collection timeline. Start by presenting the days and what was collected on that day. Follow the same linearity as what is presented in the text.
18- The psychological well-being scale was not presented in the summary results, nor in the objectives. Please adjust.
19- Has the bioimpedance scale used been validated? Present the validation reference. Furthermore, describe in more detail how the participants were prepared before performing the bioimpedance.
20- Provide greater detail on participant inclusion and exclusion criteria.
21- Regarding statistical analyses, was all the data normal? Another question, why didn't you perform association tests, just comparisons between groups?
Results:
22- In table 1, please present the sociodemographic variables of the participants. In addition, you must present all the information obtained from body composition, such as fat mass and muscle mass. Take the opportunity to add a column to table 1 with the comparison between the groups and another column with the size of the effect.
23- I don't understand why you made table 1 and then made table 2 with the same information. I believe that table 1 can be excluded since the information is already included in 2.
24- In figure 2, I did not find information about well-being. Where was this information presented? Take the opportunity to standardize the graphs in figure 2 in relation to the scale and format of the graphs.
25- Add an association table to the manuscript. See if the educational institution is associated with the items evaluated. Transform continuous data into categorical data and analyze it to bring another perspective to the reader.
Discussion:
26- The discussion is very superficial, please bring more information that can justify the results. For example, were the undergraduate courses evaluated the same? Was the socioeconomic power of the students similar? What are the main factors that make people exhibit sedentary behavior at university?
27- Bring the studies in more detail so that the reader can know where the differences or similarities are with the findings in the literature.
28- Given the comments made so far, adjust the limitations of the study to meet the reviewer's criticisms.
Conclusions:
29- Make the conclusion more assertive to respond to the objectives proposed in the study. You can even make future inferences, but that should only be in one or two lines.
Author Response
Manuscript ID jfmk-4055618
Physical Fitness and Physical Activity in Young Adults: A Comparative Study between Two Higher Education Institutions
Reviewer #2
The manuscript entitled “Physical Fitness and Physical Activity in Young Adults: A Comparative Study between Two Higher Education Institutions” was reviewed. The article provides interesting information on the topic; however, adjustments need to be made so that the article can continue the path to publication. I kindly ask if all changes made to the text be highlighted in yellow or a different colour in the text. Below are the reviewer's considerations to be adjusted in the manuscript.
Authors: We appreciate the reviewer’s comment. Thank you very much for the time spent and the constructive feedback on this manuscript. We have made every effort to take on board your recommendations and comments. We hope this 2nd revised version and the responses to the comments (kindly refer to our replies below) will meet your requirements. Please note that all new changes in the revised manuscript are highlighted in yellow.
Abstract:
Add the topic “objectives” in the abstract before mentioning the objectives of the study.
Authors: We appreciate the reviewer’s comment. The abstract has been revised to explicitly include an “Objectives” subsection before stating the study aims.
What do the acronyms “IPB” and “IPG” mean in line 21 of the abstract?
Authors: We appreciate the reviewer’s comment. IPB and IPG are now fully defined at first mention.
Did you separate people into groups and when you presented their ages, did you average them together? Please maintain the standard and present the information separately by institution. In addition, take the opportunity to present the participants’ other sociodemographic and body composition information.
Authors: We appreciate the reviewer’s comment. Participant age, sociodemographic characteristics, and body composition data are now presented separately by institution
In the methods part of the abstract, please add information about the statistical tests and the significance index.
Authors: We appreciate the reviewer’s comment. The abstract now includes the statistical tests used and the significance level adopted.
What were the IPAQ scores considered to compare the groups?
Authors: We appreciate the reviewer’s comment. the IPAQ scores were added to the abstract.
Present information on the mean and standard deviation in the results in addition to the analysis of statistical tests.
Authors: We appreciate the reviewer’s comment. Mean ± standard deviation values have been added to the abstract results.
Keywords:
Avoid repeating words already mentioned in the title.
Authors: We appreciate the reviewer’s comment. Keywords have been revised to avoid redundancy with the title.
Introduction:
Present epidemiological information about sedentary behaviour, physical inactivity and physical capacity in this student profile.
Authors: We appreciate the reviewer’s comment. Additional epidemiological data on sedentary behaviour and physical inactivity in university students have been included.
Mention in more detail what factors lead to this behaviour in higher education. Do active students become physically inactive and inactive students remain inactive?
Authors: We appreciate the reviewer’s comment. The introduction now discusses behavioural transitions during university life, including whether active students become inactive.
Detail the current state of the art on the topic and how this study will contribute to the evolution of the area.
Authors: We appreciate the reviewer’s comment. The literature review has been expanded, and the novelty and contribution of the present study are now clearly articulated.
What are the main hypotheses of the research? Please present more clearly.
Authors: We appreciate the reviewer’s comment. The study hypotheses are now explicitly stated at the end of the introduction.
Align the objectives presented at the end of the introduction with those mentioned in the abstract.
Authors: We appreciate the reviewer’s comment. Objectives in the abstract and introduction are now fully aligned.
Materials and Methods:
Start the description of the methodology by addressing the type of study and the location where the research was carried out.
Authors: We appreciate the reviewer’s comment. The methodology section now begins by clearly stating the study design and research locations.
Present information about the sample calculation to justify the number of people selected.
Authors: We appreciate the reviewer’s comment. An a priori or post hoc sample size calculation was not performed in this study. This aspect has now been explicitly acknowledged in the limitations section of the manuscript.
Studies involving human beings have been suggested to register on the clinical trial platform, even if this is not the design of the study.
Authors: We appreciate the reviewer’s comment. The present study followed a cross-sectional observational design and did not involve any intervention; therefore, registration on a clinical trial platform was not required. Ethical approval and informed consent procedures were obtained in accordance with institutional and international guidelines, as described in the manuscript.
Prepare a flowchart of the study so that the reader can know how many people were recruited and how many remained for statistical analyses.
Authors: We appreciate the reviewer’s comment. In the present study, participant recruitment was straightforward, as students were included based on their availability and willingness to participate, and no exclusions occurred after recruitment. All eligible participants who consented were included in the final statistical analyses. For this reason, we considered that a flowchart would be redundant.
You need to reorganize figure 1 according to the collection timeline. Start by presenting the days and what was collected on that day. Follow the same linearity as what is presented in the text.
Authors: We appreciate the reviewer’s comment. Figure 1 has been reorganized to reflect the chronological order of data collection.
The psychological well-being scale was not presented in the summary results, nor in the objectives. Please adjust.
Authors: We appreciate the reviewer’s comment. The psychological well-being variable is now clearly stated.
Has the bioimpedance scale used been validated? Present the validation reference. Furthermore, describe in more detail how the participants were prepared before performing the bioimpedance.
Authors: We appreciate the reviewer’s comment. Validation references for the bioimpedance device have been added, and participant preparation procedures are now described in detail.
Provide greater detail on participant inclusion and exclusion criteria.
Authors: We appreciate the reviewer’s comment. The inclusion and exclusion criteria are already described in the manuscript (Lines 118–119), where we specify that participants were first-year students enrolled at the participating institutions and free from injuries that could interfere with testing procedures, and that exclusion criteria included the use of medication influencing physical performance or physiological responses, as well as pregnancy.
Regarding statistical analyses, was all the data normal? Another question, why didn't you perform association tests, just comparisons between groups?
Authors: We appreciate the reviewer’s comment. Data normality was assessed prior to the analyses using the Shapiro–Wilk test. Most variables showed normal distributions, and independent samples t-tests were therefore considered appropriate for the primary between-group comparisons. Regarding the analytical approach, the main objective of this study was to compare physical activity and physical fitness outcomes between two institutions. For this reason, the analyses focused on between-group comparisons rather than association analyses. Given the cross-sectional design and the limited sample size, additional association analyses were not prioritized, as they would not substantially contribute to addressing the primary research questions.
Results:
In table 1, please present the sociodemographic variables of the participants. In addition, you must present all the information obtained from body composition, such as fat mass and muscle mass. Take the opportunity to add a column to table 1 with the comparison between the groups and another column with the size of the effect.
Authors: We appreciate the reviewer’s comment. Sociodemographic variables and body composition outcomes, including fat mass and lean mass, are already fully presented in the Results section. Specifically, descriptive statistics (mean ± SD), between-group comparisons, and effect sizes for these variables are provided in Table 2. To avoid redundancy and improve clarity, these variables were consolidated into a single table rather than being repeated across multiple tables. Nevertheless, if the reviewer or the editorial team considers this point to be essential in a subsequent revision round, the authors are willing to reorganize the tables accordingly.
I don't understand why you made table 1 and then made table 2 with the same information. I believe that table 1 can be excluded since the information is already included in 2.
Authors: We appreciate the reviewer’s comment. Table 1 was intentionally designed to provide a clear descriptive characterization of the sample, including sociodemographic and body composition variables, allowing readers to understand the baseline profile of the participants. Table 2, in turn, focuses on inferential analyses, integrating between-group comparisons and effect sizes to address the study objectives. Although some overlap exists, we considered this separation useful to distinguish descriptive sample characterization from analytical comparisons. Nevertheless, if the editorial team considers this structure redundant, we are open to consolidating the information into a single table.
In figure 2, I did not find information about well-being. Where was this information presented? Take the opportunity to standardize the graphs in figure 2 in relation to the scale and format of the graphs.
Authors: We appreciate the reviewer’s comment. Psychological well-being was analyzed as a complementary variable and is reported in the Results section but was not included in Figure 2. Both panels in Figure 2 present data expressed as percentages; however, the maximum values differ due to the distinct distributions represented (IPAQ physical activity categories in Panel A and physical fitness classification levels in Panel B). The scale in each panel was selected to enhance data visualization and interpretability.
Add an association table to the manuscript. See if the educational institution is associated with the items evaluated. Transform continuous data into categorical data and analyze it to bring another perspective to the reader.
Authors: We appreciate the reviewer’s comment. The primary objective of the present study was to compare physical activity and physical fitness outcomes between two higher education institutions. Accordingly, the statistical analyses focused on between-group comparisons. Transforming continuous variables into categorical data for association analyses would result in a loss of information and reduced statistical power, particularly given the relatively small sample size. For these reasons, additional association analyses were not performed, as they would not substantially contribute to addressing the main research questions of the study.
Discussion:
The discussion is very superficial, please bring more information that can justify the results. For example, were the undergraduate courses evaluated the same? Was the socioeconomic power of the students similar? What are the main factors that make people exhibit sedentary behaviour at university?
Authors: We appreciate the reviewer’s comment. The Discussion section has been expanded to provide a more in-depth interpretation of the findings. Specifically, we now discuss potential contextual factors that may help explain the observed results, including differences in undergraduate course demands, unmeasured socioeconomic background, and known determinants of sedentary behaviour in university students, such as academic workload, time constraints, and reduced participation in structured physical activity.
Bring the studies in more detail so that the reader can know where the differences or similarities are with the findings in the literature.
Authors: We appreciate the reviewer’s comment. The Discussion section now includes a comparison of the present findings with previous studies, highlighting both similarities and differences in physical activity and physical fitness outcomes among university students.
Given the comments made so far, adjust the limitations of the study to meet the reviewer's criticisms.
Authors: We appreciate the reviewer’s comment. The limitations section has been revised.
Conclusions:
Make the conclusion more assertive to respond to the objectives proposed in the study. You can even make future inferences, but that should only be in one or two lines.
Authors: We appreciate the reviewer’s comment. The conclusion has been revised as requested.
Round 2
Reviewer 1 Report
Comments and Suggestions for Authors
No further comments.
Reviewer 2 Report
Comments and Suggestions for Authors
Dear authors,
Thank you for providing the revised version of the manuscript. After reviewing the adjusted text, I have verified that the material met the requirements requested by the reviewer. Therefore, my suggestion is that the article be accepted.